# Hydrodynamic Delivery: Characteristics, Applications, and Technological Advances

**DOI:** 10.3390/pharmaceutics15041111

**Published:** 2023-03-31

**Authors:** Takeshi Suda, Takeshi Yokoo, Tsutomu Kanefuji, Kenya Kamimura, Guisheng Zhang, Dexi Liu

**Affiliations:** 1Department of Gastroenterology and Hepatology, Uonuma Institute of Community Medicine, Niigata University Medical and Dental Hospital, Minamiuonuma 949-7302, Niigata, Japan; 2Department of Preemptive Medicine for Digestive Diseases and Healthy Active Life, School of Medicine, Niigata University, Niigata 951-8510, Niigata, Japan; 3Department of Gastroenterology and Hepatology, Tsubame Rosai Hospital, Tsubame 959-1228, Niigata, Japan; 4Department of General Medicine, School of Medicine, Niigata University, Niigata 951-8510, Niigata, Japan; 5Department of Pharmaceutical and Biomedical Sciences, College of Pharmacy, University of Georgia, Athens, GA 30602, USA

**Keywords:** hydrodynamic injection, systemic, regional, capillary

## Abstract

The principle of hydrodynamic delivery was initially used to develop a method for the delivery of plasmids into mouse hepatocytes through tail vein injection and has been expanded for use in the delivery of various biologically active materials to cells in various organs in a variety of animal species through systemic or local injection, resulting in significant advances in new applications and technological development. The development of regional hydrodynamic delivery directly supports successful gene delivery in large animals, including humans. This review summarizes the fundamentals of hydrodynamic delivery and the progress that has been made in its application. Recent progress in this field offers tantalizing prospects for the development of a new generation of technologies for broader application of hydrodynamic delivery.

## 1. Introduction

Hydrodynamic delivery was established in 1999 as a simple and efficient nonviral method for the delivery of plasmids to hepatocytes in mice [1,2]. Because injection of plasmids in saline containing no other components only weakly activates host immunity, while such activation is problematic with other delivery methods [3,4,5], applications of hydrodynamic delivery in the gene and cell therapy field have been broadly explored. Significant initial efforts have been made to determine the underlying mechanisms of hydrodynamic delivery and to develop a modified procedure that is applicable to large animals and suitable for clinical use. This paper summarizes recent progress toward the successful use of hydrodynamic delivery for research and clinical applications. Our objective is to inspire innovations that extend the use of hydrodynamic delivery from genes to other substances, such as oligonucleotides, proteins, small molecules, and even genome editing machinery, and from use in animals to clinical applications. In the first half, key determinants of hydrodynamic delivery were explained along with comprehensive observations through the course of hydrodynamic delivery. Thereafter, studies utilizing hydrodynamic delivery, especially in the last five years, are collectively presented from the view of several aspects: targeted animal species and routes, types of diseases, and delivering materials/strategies.

## 2. Characteristics of Hydrodynamic Delivery

A single injection of less than 50 µg of plasmid DNA in saline through a mouse tail vein over a period of 5 s in a volume equal to 8 to 10% of the animal’s body weight results in transgene expression in up to 40% of hepatocytes [1]. A key determinant of the efficiency of hydrodynamic delivery is the anatomical structure and the expansion rate of the target organs after intravascular injection. A rapid influx of a large amount of solution into a capillary quickly extends the cell membrane and creates an invagination through which the solution enters the cell interior [6]. Our previous work employing computed tomography and contrast medium showed that the optimal expansion rate for the liver is 60%/5 s in mice [7].

The successful delivery of compounds, such as dyes, low-molecular-weight compounds, proteins, bacterial artificial chromosomes over 100 kbp in size, and even particles approximately 6 to 8 µm in size [8,9,10], by hydrodynamic delivery suggests that hydrodynamic delivery is not a receptor-mediated process [6,11]. Rapid elimination of pre-expressed green fluorescent protein in the cytosol using hydrodynamic injection [11], quick recovery from plasma alanine aminotransferase surge after the procedure [12], transgene expression in cultures of hepatocytes isolated soon after hydrodynamic delivery [11], and the existence of a transient window after hydrodynamic delivery during which nucleic acids traverse the cell membrane in the absence of hydrodynamic force [11,13] are other bricks in the wall, supporting the idea that hydrodynamic delivery generates transient pores or membrane defects in hepatocytes that facilitate intracellular delivery.

Capillaries connect arteries and veins and can be divided into three classes based on differences in two components, the endothelium and the basement membrane [14] (Figure 1). Continuous capillaries that consist of tightly connected endothelium and basement membrane without gaps prevent the leakage of water-soluble materials of 1 kDa or larger in size and exist abundantly in the body in the muscle, skin, lungs, connective tissue, and the central nervous system. Sinusoid capillaries, which are also called discontinuous capillaries, provide large interendothelial gaps over 1 µm in size in which there are incomplete shielding by the basement membrane, allowing molecules 100 kDa or larger in size to readily transude. Leaky capillaries of this type are present in organs, such as the liver, spleen, lymph nodes, and bone marrow. The third type of capillary is the fenestrated capillary; in these capillaries, small fenestrae of 50–80 nm are present in the endothelium, which has a complete basement membrane and is a component of the kidneys, small intestine, pancreas, endocrine glands, and so on. Therefore, organs in which sinusoid capillaries are present are the most suitable targets for hydrodynamic delivery.

Among the organs that contain sinusoid capillaries, direct connections with the inferior vena cava and a unique system of the portal vein make the liver an ideal target for hydrodynamic delivery from the tail vein. A large volume of solution rapidly injected into the tail vein travels to the heart and induces cardiac congestion, followed by rapid retrograde flow into the hepatic veins, which directly transfers the hydrodynamic impact to the liver [7]. The specific infrastructure of the portal vein enables steady blood flow from the intestines to the liver; this accounts for 70 to 80% of the influx to the liver, along with residual inflow via the hepatic artery. This natural portal flow counteracts the hydrodynamic retrograde flow and inhibits spillover of the injected solution into the portal vein. At the same time, the portal system provides extra space that can accommodate pushed-back preexisting blood and remove nucleases from sinusoids.

Although rapid injection of a small volume of fluid can induce a high pressure comparable to that caused by injection of a volume corresponding to 8 to 10% of body weight if the injection speed is high enough, the delivery efficiency achieved is not equivalent to that obtained using authentic hydrodynamic delivery [1]. In a fibrotic liver, injection under the standard hydrodynamic conditions gives rise to much higher pressure and stronger shear stress than those reached using the same injection profile in a normal liver, but transgene expression is markedly lower [15,16]. Slow injection of a large volume over a longer period can cause the liver to expand to a size similar to that resulting from hydrodynamic delivery; however, gene delivery occurs with much lower efficiency [7]. These observations clearly indicate that the static pressure, shear stress, and expanded volume of the liver are not the sole determinants of the efficiency of hydrodynamic delivery.

The liver lobes consist of microscopic units (lobules) that are hexagonal in shape and in which the central structure is a terminal hepatic venule of the central vein; the peripheral vertices are delineated by portal tracts consisting of the portal vein, the hepatic artery, and the bile duct. Because the central vein and portal tracts contain structures with relatively higher rigidity, such as the basal membrane and vascular smooth muscle, the intervening parenchyma is the most vulnerable to physical stretch [17]. The middle zone between the central vein and portal tracts consists of a single hepatocyte layer facing the sinusoids. When rapid flow enters the liver from the central vein, it passes through the sinusoids in the middle zone and exits into the portal veins or *vice versa*. Mechanistically, it is predictable that a rapid flow that exits from a rigid inlet toward a rigid outlet would accumulate most at the front of the rigid outlet. Transgene expression has been found to occur mainly at the end of the middle zone opposite the injection site when hydrodynamic delivery is performed either from the inferior vena cava or the portal vein [18].

For efficient hydrodynamic delivery, the physical impact of the movement of the solution must be transferred to target cells through movement of the solution at a sufficient speed after it travels from the injection site to the target organ through vasculature consisting of endothelium and basement membrane. The anatomical structure, large cell surface area facing the vasculature, and high capacity for gene expression are the primary reasons that an extremely high level of gene expression is achievable in the liver via hydrodynamic tail vein injection in mice. If a physical impact that can traverse the endothelium and basement membrane and cause organs to expand quickly is achievable through local regional injection, hydrodynamic delivery could be a promising strategy, not only for the liver but also for other organs. In fact, a wide variety of organs have been targeted using hydrodynamic delivery through the regional vasculature of target organs.

## 3. Applications of Hydrodynamic Delivery

### 3.1. Target Animals, Organs, and Routes of Injection

Hydrodynamic delivery has been successfully used to target different organs through various routes in a variety of species (Table 1). Mice [1,2,12] and rats [12,19] are the animals most commonly used for systemic injection from the tail vein; treeshrews are systemically injected using the retro-orbital sinus [20], and chickens using the jugular vein [21]. The liver is the primary organ affected by systemic injection, but substantial transgene expression has also been confirmed in other organs, such as the kidneys [21], brain capillary endothelial cells [22], and even extraordinary tissues, such as subcutaneously implanted colorectal cancer cells [23,24] and fetuses in pregnant animals [25].

Regional injection has been performed both in small animals, such as mice and rats, and in large animals, such as rabbits [26,27], pigs [12,28,29,30,31,32,33,34,35,36], dogs [37,38], monkeys [39,40], baboons (in preparation), and even humans [41,42]. Regional hydrodynamic delivery turns not only the vein but also other vessels into a potential route for effective delivery, although it is necessary to place outflow blockades at the corresponding vasculatures when the injection is performed in an antegrade fashion, such as through arteries or portal veins. In small animals, not only the liver but also the kidneys [12,43,44,45], muscle [12,46,47,48,49], pancreas [50], gonads [39], hepatocellular carcinoma [51,52], and brain tumors [53] have been targeted in vivo, while, to date, the liver, kidneys, and muscles are the targeted organs in large animals.

In terms of the liver, the hepatic vein [27,29,34,37,38], the portal vein [18,31,33,54], and the bile duct [28,30,55,56,57], in addition to the inferior vena cava [32,58], have been utilized. Materials have also been effectively delivered to hepatocellular carcinoma and the surrounding hepatocytes through the hepatic artery [51]. In humans, the liver and the saphenous vein were targeted in an ex vivo setting [41,42].

**Table 1 pharmaceutics-15-01111-t001:** Animals, target organs, and routes for which hydrodynamic delivery has been applied.

Target\Animal		Mouse	Rat	Treeshrew	Chicken	Rabbit	Pig	Dog	Monkey	Baboon	Human
Systemic	LVR	TV [1,2]	TV [19]	ROS [20]	JV [21]						
	KDNY				JV [21]						
	BCEC	TV [22]									
	FTS	TV [25]									
	IST HCC	TV [23,24,52]									
Regional	LVR	IVC, PV [2,12,18]	IVC, PV, BD, ex vivo [12,54,57,59]			IVC, HV [27]	IVC, HV, PV, BD [12,28,29,30,31,32,33,34,35,55,56,58]	HV [37,38]		HV under prep.	ex vivo [41]
	KDNY	RV, RP [43]	RV [12,44,45,60]				RV [12]				
	MSL	TA, LV, TV [46,47,48,61,62]	LV, LA * [12,49,63,64,65,66,67]			LV [26]	LV, LA [36,67]		LV, LA [39,40]		
	PCAS		SMV [50]								
	GND		LA, GV, GA [39]								
	HCC		HA [51]								
	BT		CA [22,53]								
	MCD		ex vivo [42,68]								
	SV										ex vivo [42]

LVR, liver; KDNY, kidney; BCEC, brain capillary endothelial cell; FTS, fetus; IST, implanted subcutaneous tumor; TV, tail vein; ROS, retro-orbital sinus; JV, jugular vein; MSL, muscle; PCAS, pancreas; GND, gonad; MCD, myocardium; SV, superior mesenteric vein; HCC, hepatocellular carcinoma; BT, brain tumor; IVC, inferior vena cava; PV, portal vein; RP, renal pelvis; RV, renal vein; LV, limb vein (dorsalis pedis vein, great saphenous vein, distal saphenous vein, brachial vein, cephalic vein, median vein, femoral vein); TA, tail artery; BD, bile duct; LA, limb artery (brachial artery, femoral artery, popliteal artery, iliac artery); SMV, superior mesenteric vein; GV, gonadal vein; GA, gonadal artery; HA, hepatic artery; CA, carotid artery; HV, hepatic vein; *, including mitochondrial delivery.

To target the muscles, the tail artery [61] or corresponding limb veins or arteries were utilized, including the brachial/popliteal arteries, the great saphenous vein [26,39,40,47,48,62], the dorsalis pedis vein [49,63,64], the cephalic vein [40], the median vein [40], and the femoral vein [65] and artery [66,67]. It was reported that injection of papaverine/histamine into the target vasculature enhances delivery efficiency in muscles by relaxing the endothelial shield [66,67]. Injection from the iliac artery or a corresponding vein/artery delivers genes not only to the corresponding muscles but also to the gonads (testes or ovaries), although transgene expression is essentially limited to nongermline cells, such as interstitial and granulosa cells [39].

For kidney targeting, injection has been performed through the renal vein [12,44] or the renal pelvis [43,60]. The superior mesenteric vein was employed when targeting the pancreas [50]. Even brain tumors have been targeted through the carotid artery [22,53]. Hydrodynamic impact can extracorporeally transfer genes into the myocardium in rats [42,68].

### 3.2. Targeted Diseases

The highest levels of gene expression reported in mice after a single hydrodynamic injection into a tail vein are 500 µg of protein per ml of serum [69] and 45 µg of cellular protein per gram of liver [1]. It was shown that this level of gene expression is sufficient to restore blood coagulation in hemophilic mice [70], to establish a mouse model of human viral infection [71,72,73,74], and to eliminate established tumors in tumor-bearing mice [51,53] (Table 2).

Hydrodynamic delivery has been most frequently utilized to address diseases of the liver, especially those caused by hepatitis B virus infection. Studies in which such diseases are addressed account for approximately 30% of the literature on hydrodynamic delivery in the last five years. Direct delivery of genetic material makes it possible to establish animal models that show persistent replication of infectious pathogens, overcoming species barriers. An easily manipulated animal model provides immense benefit to researchers attempting to understand its pathophysiology and to develop effective treatments. Hepatitis C virus [75,76], hepatitis D virus [74], influenza virus [77,78], infection with the malaria parasite [79,80], sepsis [81], liver fibrosis [82], and hypertriglyceridemia [83] are other conditions that have been explored using hydrodynamic delivery. In the case of the hepatitis B virus [84,85], the malaria parasite [79,80], and the influenza virus [77,86], hydrodynamic delivery has been employed for vaccine development.

Hydrodynamic delivery has been used to develop animal models not only of hepatitis B virus infection but also of a wide variety of other diseases, including hepatitis C virus infection [75,76,87,88,89,90,91], hepatocellular carcinoma [92,93,94,95], hepatoblastoma [96,97,98], cholangiocellular carcinoma [99,100,101,102], liver fibrosis [103,104], nonalcoholic fatty liver disease [103,105,106], hemophilia B [107], von Willebrand disease [108,109], thrombotic thrombocytopenic purpura [110,111], malaria [79,80], enterovirus 71 infection [112], pseudoxanthoma elasticum [113], psoriasis [114], intracerebral aneurysm [115], the human immune system [116,117], and metastasis of melanoma, breast cancer, and renal cell carcinoma to the lungs, liver, and kidney [118,119].

Hepatocellular carcinoma is the second most frequently explored liver disease; it is addressed in approximately 10% of the total publications employing hydrodynamic delivery in the last 5 years. Malignant diseases, including hepatoblastoma, cholangiocellular carcinoma, colorectal cancer [23,24,120,121,122], lung cancer [123], brain tumors [53], lymphoma [124], melanoma [24,121,125,126], and metastasis of renal cell carcinoma to the lungs, liver, and kidneys [118], have been investigated using hydrodynamic delivery. More than half of the reports on malignant diseases focus on hepatocellular carcinoma.

Other major target illnesses are hereditary diseases. Hemophilia A [127,128] and B [70,107,129,130] have been eagerly explored with respect to feasibility of treatment. The other inherited diseases explored include pseudoxanthoma elasticum [113], von Willebrand disease [108,109,131,132], thrombotic thrombocytopenic purpura [110,111,133,134], mucopolysaccharidosis I and VII [135,136], phenylketonuria [137], tyrosinemia [138,139,140], Leber congenital amaurosis [141], sickle cell disease [142], cystathionine β-synthase deficiency [143], Fabry disease [144], alpha-1 antitrypsin deficiency [69], growth hormone deficiency [145], metachromatic leukodystrophy [146], short-chain acyl-coA dehydrogenase deficiency [147], and muscular dystrophy [148].

Other diseases can be grouped into liver-related diseases and other diseases. Liver-related diseases include liver fibrosis/cirrhosis [149,150,151], nonalcoholic fatty liver disease [152,153], alcoholic liver injury [154,155], portal hypertension [156], fulminant hepatitis [157,158], liver regeneration [159], and acute liver injury [160,161]. Extrahepatic diseases include infectious diseases, such as bacterial [162] and trypanosome [163] infection, cutaneous diseases [117] including atopic skin [164,165], cardiovascular diseases, such as atherosclerosis [166], cardiac injury [167,168], intracranial aneurysm [115], ischemia and hypoxic-ischemic encephalopathy [169,170,171], glomerulonephritis [172], renal injury [173], renal fibrosis [174], and hyperparathyroidism [175], metabolic disorders, such as diabetes mellitus, obesity, and hypertriglyceridemia [83,176,177,178,179], and inflammatory disorders, such as myocarditis [180,181], pancreatitis [182], autoimmune disease [183], arthritis [184], osteoporosis [185], transplant rejection [59,186], xenotransplantation [186], organophosphate intoxication [187], humanized mast cells [117], and diseases of the human immune system [116].

**Table 2 pharmaceutics-15-01111-t002:** Diseases for which hydrodynamic delivery has been utilized to explore the pathogenesis and/or therapeutic potential.

Infectious	Cancer	Hereditary	Liver
Hepatitis B virus (HBV) [72,73,84,85]	Hepatocellular carcinoma [51,92,93,94,95]	Hemophilia A and B [70,107,127,128,129,130]	Liver fibrosis [82,103,104,149,150,151]
Hepatitis C virus [71,75,76,87,88,89,90,91]	Hepatoblastoma [96,97,98]	Pseudoxanthoma elasticum [113]	Nonalcoholic fatty liver diseases [103,105,106,152,153]
Hepatitis D virus [74]	Cholangiocellular carcinoma [99,100,101,102]	von Willebrand disease [108,109,131,132]	Alcoholic liver injury [154,155]
Influenza virus [77,78]	Colorectal cancer [23,24,120,121,122]	Thrombotic thrombocytopenic purpura [110,111,133,134]	Portal hypertension [156]
Enterovirus 71 [112]	Lung cancer [123]	Mucopolysaccharidosis I and VII [135,136]	Fulminant hepatitis & regeneration [157,158,159]
Vaccination (HBV, Malaria, Influenza) [77,79,80,84,85,86]	Brain tumor [53]	Phenylketonuria [137]	Acute liver injury [160,161]
Malaria parasite [79,80]	Lymphoma [124]	Tyrosinemia [138,139,140]	**Others**
Streptococcus [162]	Melanoma [24,125,126]	Leber congenital amaurosis [141]	Atopic skin & cutaneous diseases [114,117,164,165]
Sepsis [81]	Metastasis (melanoma, breast cancer, RCC * (lungs, liver, kidneys)) [118,119,121]	Sickle cell disease [142]	Cardiovascular & ischemic diseases [115,166,167,168,169,170,171]
Trypanosome [163]	*, renal cell carcinoma	Cystathionine β-synthase deficiency [143]	Kidney diseases & hyperparathyroidism [172,173,174,175]
		Fabry disease [144]	Diabetes mellitus & obesity [176,177,178,179]
		α-1 antitrypsin deficiency [69]	Hypertriglyceridemia [83]
		Growth hormone deficiency [145]	Inflammatory diseases [180,181,182,183,184]
		Metachromatic leukodystrophy [146]	Osteoporosis [185]
		Short-chain acyl-CoA dehydrogen. def. [147]	Transplantation & intoxication [59,186,187]
		Muscular dystrophy [148]	Humanized immune system [116,117]

### 3.3. Technological Issues

#### 3.3.1. Delivery of Materials

Hydrodynamic delivery is the simplest gene delivery method that has been developed; in this method, the only materials required to deliver genes are an expression vector and saline. Although no protective ingredients are included in its formulation, hydrodynamic delivery can act as an effective carrier, not only of DNA [33,114,124,132,136,137,143,171] but also of other materials, including RNA and proteins, because rapid injection of a large volume of solution physically pushes away preexisting blood components, including nucleases, and brings nucleic acids or proteins directly into the cell interior within a short period of time. Since the establishment of the concept of RNA interference to suppress gene expression, there have been reports of the effective hydrodynamic delivery of various types of RNA, including microRNA [47,150,152,188,189,190,191], circular RNA [126], short hairpin RNA [94,192,193], and small interfering RNA (siRNA) [87,194,195,196]. Hydrodynamic delivery can also extrude cells from blood vessels and, thus, offers a simple and easy way to develop animal models for metastatic diseases [118] (Table 3).

#### 3.3.2. Genome Editing (Somatic Gene Editing)

For a long period, gene delivery was performed to explore the functions of specific genes or to confirm the efficacy of treatment through transgene expression. Aiming at long-lasting gene expression using nonviral vectors and in addition to epigenetic control [227], plasmids that replicate in an episomal fashion and site-specific integration systems were developed and hydrodynamically delivered. In terms of site-specific integration, PhiC31 integrase [197], sleeping beauty transposon [30,96,111,142,191,199,200,201,202], piggyBac transposon [126,207], and Cre-loxP and technologies derived from it [208,209,210] were a major focus. Tamoxifen-dependent Cre recombinase (CreER) can initiate the recombination event at any desired time point [201,212]. In optogenetic genome engineering, split fragments of photoactivatable Cre can be hydrodynamically delivered and assembled so that the recombination process proceeds upon blue-light illumination, allowing for control of recombination events, not only in time but also in space [209].

The epoch-making discovery and establishment of the CRISPR (clustered regularly interspaced short palindromic repeats)-Cas9 system are rapidly progressing toward use in in vivo gene editing. Hydrodynamic delivery plays a key role in delivering the components required for in vivo site-directed gene editing, not only for CRISPR–Cas9 [25,95,203,218,219] but also for other systems, such as prime editors [141], split prime editors [140], adenosine deaminase acting on RNA [226], and adenine base editors [139].

#### 3.3.3. Regional Hydrodynamic Delivery

Human application is a final goal of gene delivery system development. Because hydrodynamic impacts generated by a systemic injection through the tail vein in mice are temporarily overwhelming for the cardiovascular system, hydrodynamic impacts must be limited around a target site when hydrodynamic delivery is applied in humans. The insertion of an injecting device into a corresponding vasculature to target an organ or a part of an organ is an established technique in a clinic as interventional radiology; however, reproduction of sufficient hydrodynamic impacts at a target region is challenging.

In the hydrodynamic delivery of material injected into the tail vein, the injected solution never flows out of the body. Therefore, the body is a closed system with respect to the injected solution (Figure 2). Under closed circulation, the hydrodynamic impact of the injection is reproducibly generated as a function of injection volume and speed. On the other hand, in regional hydrodynamic delivery, the solution is injected into an open system and can readily flow out of the target area through latent vascular connections [228]. Therefore, the hydrodynamic impact of regional hydrodynamic delivery cannot be reproducibly generated using fixed parameters of injection volume and speed. To achieve safety and reproducibility in the open system, we developed a computer-controlled hydrodynamic delivery system, HydroJector, in which the injected solution is propelled by carbon dioxide gas [12] or by an electric motor [198]. By compensating for leakage from the target area, the system controls the injection in a way that creates a reproducible intravascular pressure–time curve at the injection site. Local regional hydrodynamic delivery has been successfully performed by inserting a catheter into a vessel appropriate for the target organ, such as the renal vein for the kidney, the dorsalis pedis vein for muscle, and the superior mesenteric vein for the pancreas.

Ultrasound-targeted microbubble destruction has been explored as a way of enhancing hydrodynamic delivery efficiency [178]. Sound energy that is focused with a proper combination of acoustic pressure, pulse repetition frequency, and duty cycle has the potential to cause cavitation of the cell membrane. It was reported that cavitation prolongs transgene expression when coupled with hydrodynamic force [178]. Similarly, various nonviral and viral vectors, such as polyplexes [57,62,217], cationic liposomes [135], adeno-associated virus [86,103,220,221,222,223], lentivirus [38,224,225], and foamy virus [34], have been administered under hydrodynamic conditions to achieve improved gene transfer.

#### 3.3.4. Miscellaneous

There are many technological developments and/or introductions of new concepts that assist in hydrodynamic delivery. Bioluminescence imaging makes it possible to quantify transgene expression in a living animal [47,203,204,205,206]. A tissue-clearing technology that makes it possible to detect the expression and intracellular fate of transgenes within the three-dimensional architecture of tissues without a need for the preparation of tissue sections has been developed [206]. This sensitive technology was used to show that transgene expression after hydrodynamic delivery can be detected, not only in the liver but also in other tissues, including heart, kidneys, spleen, lungs, stomach, and small and large intestines, although the delivery efficiency was 4–6 orders of magnitude lower than that in the liver.

Repopulation is a strategy in which cells that carry lethal genetic abnormalities are replaced by cells that are designed to survive under lethal pressure [189,211]. Hepatocytes carrying fumarylacetoacetate hydrolase gene deficiency, a lethal genetic alteration, are rescued by deleting the hydroxyphenylpyruvate dioxygenase gene; this change blocks tyrosine catabolism and prevents fatal accumulation of toxic tyrosine catabolites. If genetic modifications coupled with hydroxyphenylpyruvate dioxygenase gene deletion are introduced, all of the hepatocytes that carry fumarylacetoacetate hydrolase gene deficiency are replaced by the modified cells due to selective survival of the modified cells. Simple selection of a target-specific single-guide RNA in the CRISPR–Cas9 system allows one to obtain a liver consisting of hepatocytes that carry the chosen genetic modification. Use of repopulation strategies may render the gene delivery efficiency of the method itself unimportant, and such strategies offer a promising therapeutic option for genetic diseases in which defective hepatocytes do not survive.

Because the liver is a preferential target of hydrodynamic delivery in both the systemic and regional approaches, there is a demand for reprogramming of hepatocytes to other cell types, such as bile duct cells [202], insulin-producing cells [178,213,214], and pluripotential cells [215,216]. Another use of reprogramming is for the reversal of malignant cells to nonmalignant cells [97].

## 4. Conclusions

Comprehension of the mechanism through which hydrodynamic delivery works enables us to overcome the biological barriers associated with the capillary endothelium and cell membranes and can lead to the development of methods for the efficient intracellular delivery of biologically active materials. The physical nature of the hydrodynamic delivery system sets no restriction on the type of material that is delivered, and the method has broad application in basic and clinical investigations of the effects of various biologically active materials, including gene-coding sequences, RNAs, oligonucleotides, proteins, and even mixtures of substances, such as DNA/single-guide RNA or ribonucleoprotein of Cas9/siRNA, for genome editing. It appears to be feasible for use in hydrodynamic-based therapy in the clinic when tight control of the hydrodynamic impact of the injection can be achieved in humans. A computer-assisted hydrodynamic injection device has been developed and could serve as a stepping stone for the development of the next generation of devices for hydrodynamic delivery. An establishment of regional hydrodynamic delivery will prove a concept of sophisticated gene therapy by providing a platform that can transfer various components required for site-directed gene editing, repopulation, and/or activation control in time and space, provoking the fewest auxiliary effects.

## Figures and Tables

**Figure 1 pharmaceutics-15-01111-f001:**
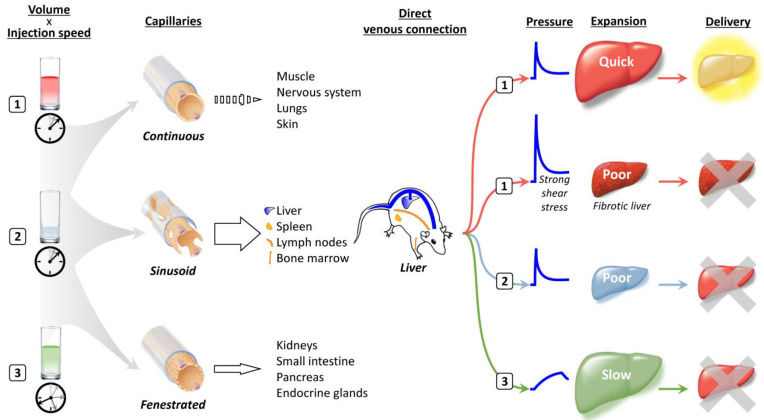
Traverse of hydrodynamic impact from injection to gene transfer sites.

**Figure 2 pharmaceutics-15-01111-f002:**
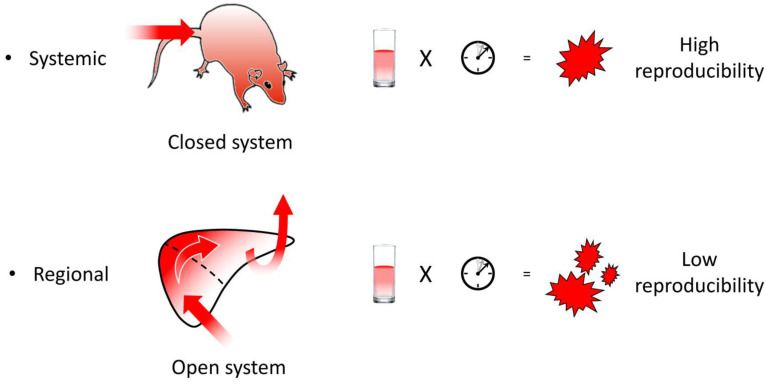
Establishment of hydrodynamic impacts in systemic and regional injections.

**Table 3 pharmaceutics-15-01111-t003:** Strategies/materials coupled with hydrodynamic delivery.

Delivery Materials	Technological Developments	Gene Editing
Minicircle DNA [33,114,124,132,136,137,143,171]	US-targeted microbubble destruction [178]	PhiC31 Integrase [197]
microRNA [47,150,152,188,189,190,191]	Computer-assisted hydrodynamic delivery [12,198]	Sleeping Beauty [30,96,111,142,191,199,200,201,202]
Circular RNA [126]	Bioluminescence imaging [47,203,204,205,206]	piggyBac [126,207]
shRNA [94,192,193]	Tissue clearing [206]	Cre-loxP [208,209,210]
siRNA [87,194,195,196]	Repopulation [189,211]	CreER [201,212]
Cell [118]	Reprogramming [178,202,213,214,215,216]	Optogenetic genome engineering [209]
Polyplex [57,62,217]		CRISPR-Cas9 [25,95,203,218,219]
Cationic liposome [135]		Prime editor [141]
Adeno-associated virus [86,103,220,221,222,223]		Split prime editor [140]
Lentivirus [38,224,225]		adenosine deaminase acting on RNA [226]
Foamy virus vector [34]		Adenine base editor [139]

## Data Availability

No new data were created or analyzed in this study. Data sharing is not applicable to this article.

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
