# Peer review of "Hydrodynamic Delivery: Characteristics, Applications, and Technological Advances"

_pharmaceutics, 2023, doi:10.3390/pharmaceutics15041111_

Round 1

Reviewer 1 Report

This review summarizes the fundamentals of hydrodynamic delivery and the progress that has been made in its application. It is a topic of interest to the researchers in the related areas. But I have some questions.

1. The high-pressure hydrodynamic method was used to establish a virus transfection mouse model. But how to achieve the control variables of pressure during the experiments?

2. The impact of blood circulation under high pressure can cause instantaneous damage to the liver of mice model of HBV hydrodynamic injection, so that the input DNA fragments can enter liver cells, but the disadvantages are short maintenance time of HBV replication and low titer. Could the authors add some comments that can reduce or avoid such disadvantages?

3. Currently, the commonly used HBV mouse models can be simply divided into five categories: transgenic mice, hydrodynamic injection models, AAV vector transfection models, human and mouse chimeric liver mouse infection models, and liver and immune system humanized mouse models. What are the advantages of hydrodynamic injection model compared with other models?

4. Many literatures are cited, but not well analyzed. The authors should focus on some key report, not just piling the papers.

5. Check and avoid typos, such “Oen” in Figure 2.

Author Response

Thank you for the reviewers’ careful reading and thoughtful comments. Along with the reviewers’ comments, the manuscript has been revised. The corrections were written in red in the revised version, and a point-by-point response to the reviewers’ critiques was listed below.

Reviewer 1

This review summarizes the fundamentals of hydrodynamic delivery and the progress that has been made in its application. It is a topic of interest to the researchers in the related areas. But I have some questions.

Q1. The high-pressure hydrodynamic method was used to establish a virus transfection mouse model. But how to achieve the control variables of pressure during the experiments?

A1. In terms of a mouse model, it is a closed system for hydrodynamic delivery through the tail vein. As long as a closed system, it is advised to keep authentic injection speed and volume for efficient hydrodynamic delivery.

Q2. The impact of blood circulation under high pressure can cause instantaneous damage to the liver of mice model of HBV hydrodynamic injection, so that the input DNA fragments can enter liver cells, but the disadvantages are short maintenance time of HBV replication and low titer. Could the authors add some comments that can reduce or avoid such disadvantages?

A2. The level of transgene expression should depend on a gene delivery method at the time of delivery, but not for the prolonged expression, which is attributable to the fate of transgene in host cells. CRISPR-cas9 and other in vivo site-directed gene editing have been coupled with hydrodynamic delivery to achieve sustained transgene effects. On the other hand, the number of cells at the time of hydrodynamic delivery is rheologically restricted to the hepatocytes along the edge of zone 2. To overcome this technological limitation, repopulation is a promising strategy. If HBV components are introduced with hydroxyphenylpyruvate dioxygenase gene deletion, all the hepatocytes that carry the fumarylacetoacetate hydrolase gene deficiency are going to be replaced by the modified cells with HBV components due to selective survival of the modified cells. Both in vivo site-directed gene editing and repopulation are discussed in the original version.

Q3. Currently, the commonly used HBV mouse models can be simply divided into five categories: transgenic mice, hydrodynamic injection models, AAV vector transfection models, human and mouse chimeric liver mouse infection models, and liver and immune system humanized mouse models. What are the advantages of hydrodynamic injection model compared with other models?

A3. The advantage of hydrodynamic delivery is the simplicity to achieve delivery efficiency sufficient for the study of the biological features of HBV. The simplicity is not only for the methodological aspect but also for the materials delivered into the cell interior, which leads to provoking the least auxiliary effects.

Q4. Many literatures are cited, but not well analyzed. The authors should focus on some key report, not just piling the papers.

A4. Our focus on citing many pieces of literature is not to unravel the subjects tackled in each manuscript, but to inspire novel applications of hydrodynamic delivery. For this purpose, the manuscripts especially in a recent-five-years are collectively summarized from the view of various aspects of hydrodynamic delivery: targeted animal species and routes, types of diseases, and delivering materials/strategies. Along with the reviewer’s comment, 10 citations have been deleted in the revised version.

Q5. Check and avoid typos, such “Oen” in Figure 2.

A5. Thank you for your careful reading. We corrected “Oen” to “Open”.

Reviewer 2 Report

The paper provides a revision of hydrodynamic delivery method. While the characteristics, applications and technological advances are listed and many papers are cited, I believe the paper currently is missing a clear aim and a strong conclusion. I suggest the authors to make it clear how their paper is adding to the field. 

Author Response

Thank you for the reviewers’ careful reading and thoughtful comments. Along with the reviewers’ comments, the manuscript has been revised. The corrections were written in red in the revised version, and a point-by-point response to the reviewers’ critiques was listed below.

Reviewer 2

Q1. The paper provides a revision of hydrodynamic delivery method. While the characteristics, applications and technological advances are listed and many papers are cited, I believe the paper currently is missing a clear aim and a strong conclusion. I suggest the authors to make it clear how their paper is adding to the field.

A1. In this review, key determinants of hydrodynamic delivery were presented based on the comprehensive observations through the course of hydrodynamic delivery. Thereafter, manuscripts, in which hydrodynamic delivery was utilized, especially in a recent-five-years are collectively summarized. This review aims to help readers make further methodological development and application of hydrodynamic delivery. Basic knowledge is inevitable for the development of a novel technology. To inspire readers with a variety of possible applications, the references are presented from the view of several aspects: targeted animal species and routes, types of diseases, and delivering materials/strategies. To clear our aim, the following description was added in the introduction section.

“In the first half, key determinants of hydrodynamic delivery were explained along with comprehensive observations through the course of hydrodynamic delivery. Thereafter, literature utilizing hydrodynamic delivery, especially in the recent-five-years are collectively presented from the view of several aspects: targeted animal species and routes, types of diseases, and delivering materials/strategies.”

Reviewer 3 Report

The review article entitled “Hydrodynamic Delivery: Characteristics, applications, and technological advances” by Suda et al. discusses the characteristics and applications of hydrodynamic delivery. This well-written manuscript could be accepted for publication in IJMS after minor revision.

My primary concerns are:

1.       The addition of a separate section discussing the challenges of hydrodynamic delivery, particularly in humans, is necessary to complete the story.

2.       Tables 1, 2, and 3 should include the critical outcomes of each study.

3.       The authors should incorporate a couple more figures.

4.       Similarly, a section describing the prospect of hydrodynamic delivery is needed.

Author Response

Thank you for the reviewers’ careful reading and thoughtful comments. Along with the reviewers’ comments, the manuscript has been revised. The corrections were written in red in the revised version, and a point-by-point response to the reviewers’ critiques was listed below.

Reviewer 3

The review article entitled “Hydrodynamic Delivery: Characteristics, applications, and technological advances” by Suda et al. discusses the characteristics and applications of hydrodynamic delivery. This well-written manuscript could be accepted for publication in IJMS after minor revision.

My primary concerns are:

Q1. The addition of a separate section discussing the challenges of hydrodynamic delivery, particularly in humans, is necessary to complete the story.

A1. A regional application is key to realizing hydrodynamic delivery in humans. To indicate this dependency, the following description was added to the “3.3.3. Regional Hydrodynamic Delivery” section in the revised version.

“A human application is a final goal of gene delivery system development. Because hydrodynamic impacts generated by a systemic injection through the tail vein in mice are temporarily overwhelming for the cardiovascular system, hydrodynamic impacts must be limited around a target site when hydrodynamic delivery is applied in humans. An insertion of injecting device into a corresponding vasculature to target an organ or a part of an organ is an established technique in a clinic as interventional radiology; however, a reproduction of sufficient hydrodynamic impacts at a target region is challenging.”

Q2. Tables 1, 2, and 3 should include the critical outcomes of each study.

A2. Our focus on citing many pieces of literature is not to unravel the subjects tackled in each manuscript, but to inspire novel applications of hydrodynamic delivery. The references except for the technological development of hydrodynamic delivery, there are no conclusive results concerning hydrodynamic delivery. In these manuscripts, hydrodynamic delivery is simply utilized to achieve each goal independently from hydrodynamic delivery. These critical outcomes of each study are beyond the scope of this review. Instead, the manuscripts especially in a recent-five-years are collectively summarized from the view of various aspects of hydrodynamic delivery: targeted animal species and routes, types of diseases, and delivering materials/strategies.

Q3. The authors should incorporate a couple more figures.

A3. Could you specifically indicate which type of figures you have in mind?

Q4. Similarly, a section describing the prospect of hydrodynamic delivery is needed.

A4. As is described in the conclusions section, the prospect of hydrodynamic delivery should be the development of a computer-assisted hydrodynamic injection device that enables tight control of the hydrodynamic impact in humans. To make this point much clear, the following descriptions were added in the Conclusions section of the revised version.

“An establishment of regional hydrodynamic delivery will prove a concept of sophisticated gene therapy by providing a platform, which can transfer various components required for a site-directed gene editing, repopulation, and/or activation control in time and space, with provoking the least auxiliary effects.”.

Round 2

Reviewer 1 Report

I've no comments now.

Reviewer 2 Report

No further comments